# Elaboration and Characterization of a New Heavy Metal Sensor Functionalized by Extracellular Polymeric Substances Isolated from a Tunisian Thermophilic Microalga Strain *Graesiella* sp.

**DOI:** 10.3390/s23020803

**Published:** 2023-01-10

**Authors:** Wejdene Gongi, Maxence Rube, Hafedh Ben Ouada, Hatem Ben Ouada, Ollivier Tamarin, Corinne Dejous

**Affiliations:** 1UMR 228 Espace-Dev, University of French Guiana, F-97300 Cayenne, France; 2Laboratory of Blue Biotechnology & Aquatic Bioproducts, National Institute of Marine Sciences and Technologies, Monastir 5000, Tunisia; 3Laboratoire des Interfaces et Matériaux Avancés, Faculté des Sciences de Monastir, Monastir University, Monastir 5000, Tunisia; 4IMS, University of Bordeaux, CNRS, Bordeaux INP, UMR 5218, F-33405 Talence, France

**Keywords:** extracellular polymeric substances, electrochemical impedance spectroscopy, acoustic wave, sensor, heavy metals

## Abstract

The present study aimed to develop and characterize new heavy metal sensors functionalized by extracellular polymeric substances (EPSs) isolated from a Tunisian thermophilic microalga strain *Graesiella* sp. The elaborated sensor showed a highly homogeneous character and revealed a microstructural lamellar arrangement, high crystalline nature, and several functional groups. Electrochemical impedance spectroscopy (EIS) and acoustic wave sensing were used as sensing techniques to explore the ability of microalgae-EPS-functionalized sensors to detect cadmium and mercury as heavy metals. For impedimetric measurements, a two-dipole circuit was adopted and showed good-fitted results with a low total error. The acoustic sensor platforms showed good compatibility with EPS in adjacent water. For both EPS-functionalized sensors, metal ions (Cd^2+^, Hg^2+^) were successfully detected in the concentration range from 10^−10^ M to 10^−4^ M. Impedimetric sensor was more sensitive to Cd^2+^ at low concentrations before saturation at 10^−7^ M, while the acoustic sensor exhibited more sensitivity to Hg^2+^ over the full range. The results highlight a new potential alternative to use microalgae EPSs as a sensitive coating material for the detection of heavy metals. However, its use in a real liquid medium requires further investigation of its selectivity in the presence of other compounds.

## 1. Introduction

Contamination of land and water remains a serious environmental issue since a large mass of toxic substances, such as heavy metals, is being released into the environment by both natural and anthropogenic sources. Heavy metals are inorganic compounds that persist for centuries in ecosystems since they are nonbiodegradable and have been proven to accumulate in living beings, thus affecting the reproductive, neurological, and immunological systems of both humans and animals [1].

Many techniques are used for heavy metal quantification but are often complex and expensive, with other intrinsic issues such as long steps of preconcentration and analysis [2,3]. In recent years, the development of biosensors has gained increasing interest due to their high sensitivity, selectivity, and accuracy [4]. The development of whole-cell and cell-free biosensors for the detection of heavy metals has raised increased interest. Microalgae such as *Arthrospira platensis* and *Chlorella Vulgaris* were used in various studies to develop whole-cell biosensors for the control of toxic pollutants in aquatic environments [5,6,7]. Immobilized microalgae cells were coated on sensor electrodes by alternating deposition of polyelectrolyte multilayers using layer-by-layer (LBL) deposit methods [5,8].

Several microalga strains showed their ability to bind several heavy metals [9]. This ability was attributed to extracellular polymeric substances (EPSs), also called exopolysaccharides, which are released by several microalgae into the surrounding environment [10,11,12]. The richness of microalgae EPSs in uronic acid and sulfate groups gives them negative surface charges [13], favoring the complexation of positively charged metal ions. Various functional groups (carbonyl, carboxyl, and hydroxyl) and protein substituents are also involved in this complexation process [14,15]. Due to their structural complexity containing hydrophilic and hydrophobic groups, EPSs can absorb and retain water, which gives them gelling properties, increasing their ability to adsorb various pollutants by simple inclusion [13].

In addition, EPSs extracted from certain microalgae showed adhesive properties [15] and pseudoplastic flow [16], which are advantageous characteristics for biosensor applications, especially when mixed with other materials. In our previous work, cyanobacteria EPS was used as a monolayer coating material for gold sensors, and successful detection of microplastics in water was performed [10].

In previous work, results showed that Chlorophyta *Graesiella* sp. cultured under controlled laboratory conditions or natural temperature and light conditions released high amounts of EPSs into the culture medium. These EPSs, characterized as heterosulfated polysaccharides composed mainly of polysaccharides (80%) and proteins (14%), presented a high crystalline and anionic nature with high emulsifying, flocculating, and film-forming properties [17,18].

In this work, we combined the advantages of the physical and chemical characteristics of *Graesiella* sp. EPS, the simplicity of their extraction, and their deposition method to elaborate new heavy metal sensors. The proposed functionalized sensors membrane was characterized using FTIR, AFM, X-ray, and HPSEC technics. Electrochemical impedance spectroscopy and acoustic wave techniques were used to study the biosensor’s analytical performances for detecting heavy metal ions, particularly Hg^2+^ and Cd^2+^.

Thus, the objective of this paper, before further investigation for a selective sensor in a real liquid medium, consisted of exploring the use of microalgae EPS as sensitive bioreceptors for heavy metals.

## 2. Materials and Methods

### 2.1. Extraction and Elaboration of the EPS-Membrane-Forming Solution

Extracellular polymeric substances (EPSs) were obtained from the cultivation of *Graesiella* sp., as mentioned in previous work [9,16].

The membrane-forming solution was obtained, as described by Gongi et al. [9], by dissolving 1 mg of lyophilized EPS in 1 mL ethanol solution (99% purity, purchased from Sigma Aldrich, St. Louis, MO, USA). The obtained solution before its deposition was characterized using zeta potential for its electrical potential and high-pressure size exclusion chromatography for its molecular weight. The zeta potential measurements were performed in triplicate at 25 ± 1 °C by measuring the dynamic electrophoretic mobility of the water-dispersed particles. All measured electrophoretic mobilities were converted into zeta potential using Smoluchowski’s formula with a sizer Nano ZSP (Malvern) [19].

The molecular weight of the deposited EPS solution (coating solution) was analyzed by high-performance size exclusion chromatography (HPSEC) with a refractive index detector. The solution was filtered through a 0.22 µm filter (Sartorius, Bohemia, New york, USA) before injection. A Shodex OH-Pak SB-805 column following an OH-Pak SB-G guard column (8 mm × 300 mm, Japan) was used at 25 °C, and the column was eluted with phosphate buffer (50 mmol L^−1^) at a flow rate of 0.8 mL min^−1^. Then, the injection volume was 20 µL. Standard dextran (0.3 g L^−1^; Sigma 31,390) of molecular weights 5, 13.5, 500, 1500, 30,000, and 60,000 kg mol^−1^ were used to build a calibration curve. The molecular mass of the EPS sample was extrapolated according to the calibration curve of standard dextran.

### 2.2. Sensor’s Elaboration

The ability of microalgal EPS as a sensitive membrane was investigated using two types of sensors (Figure 1): a gold electrode for electrochemical detection and a Love wave sensor for acoustic detection.

The gold electrode used for electrochemical detection consisted of (100)-oriented, p-type (3–5 Ω cm) silicon wafers of thickness 1 mm, thermally oxidized (800 nm thick silicon oxide layer), coated with titanium (adhesion layer, 30 nm thick) and gold (300 nm thick) deposited by evaporation under vacuum, then cut into 1.2 × 1.2 cm^2^ squares.

The Love wave sensor (500 µm thick) used for acoustic detection consists of a dual delay line on a piezoelectric substrate (quartz) covered by a rigid overlay (SiO_2_) acting as a guiding layer with interdigitated transducers (IDTs) to generate and receive the acoustic wave [5].

Both types of devices were provided by the “Laboratoire d’Analyse et d’Architecture des Systèmes” (LAAS, CNRS, Toulouse, France). The functionalization of the sensors was performed by deposition of the EPS solution (1 mg EPS mL^−1^ of ethanol) using a self-assembled monolayer technique. The immobilization of the sensitive membrane was performed by spin-coating on the cleaned gold surface electrode for electrochemical detection and by drop-casting on the Love wave sensor surface for acoustic detection. Both sensors were dried for 24 h in an oven (30 °C). The thickness values, determined using a surface profiler (Alpha-Step IQ), were 10 nm for the EPS-functionalized gold sensor and 22 nm for the EPS-functionalized Love wave sensor. These values are within the range of biological sensors (5 to 50 nm [20]).

### 2.3. Surface Characterization of the Sensitive Membrane

The structural analysis of the sensitive membrane was performed by Fourier transform infrared (FTIR) and X-ray diffraction (XRD) spectroscopies and surface microstructure characterization by atomic force microscopy (AFM) and scanning electron microscopy (SEM).

FTIR analyses were carried out using an attenuated total reflectance Fourier transform infrared spectroscopy (ATR-FTIR Perkin-Elmer). The spectra were recorded at room temperature in the range of 500–4000 cm^−1^ with a resolution of 2 cm^−1^. The Spectrum Suite ES software was used for FTIR data treatment.

Diffraction data on both gold electrode and Love wave sensors were collected with an X-ray diffractometer CICECO Empyrean (JDX 3532; PANanalytical, Hong Kong, Japan). Data analysis was performed using CrysAl-isPro Software System. The angle range of diffraction was observed from 0 to 50°.

The surface topography of the EPS-functionalized Love wave sensor was assessed with AFM (Figure 2). The AFM measurements were carried out using a Bruker Innova atomic force microscope at a frequency of ~300 kHz. Images were analyzed using the software Gwyddion-64.

The top view and cross-section morphology of the sensor membrane were inspected by scanning electron morphology SEM with an HR-FESEM SU-70 Hitachi microscope (Prior Scientific, Rockland, MA. 02370, U.S.A). First, the sensor was cryofractured by immersion in liquid nitrogen and fixed on the SEM support using double-sided adhesive tape, and then the sensor was coated with 5 nm thick gold and observed under an accelerating voltage of 5.0 kV and absolute pressure of 60 Pa.

### 2.4. Electrochemical Impedance Spectroscopy “EIS” Measurements

Electrochemical impedance spectroscopy measurements were carried out at room temperature (20 ± 3 °C) in an electrochemical cell connected to a computer-controlled impedance analyzer (FRA32M, Autolab, Metrohm, Herisau, Switzerland). The electrochemical impedance spectroscopy measurements were performed in ammonium acetate (0.04 M, pH 6.8), used as a background electrolyte, in a conventional electrochemical cell containing a three-electrode system, ensuring the electrode’s stable positioning and the solution’s stirring. EPS-coated gold electrode (0.125 cm^2^) was the working electrode, a platinum plate (0.282 cm^2^) was the counter electrode, and a saturated electrode (Ag/AgCl/KCl) served as the reference electrode (Figure 3). The amplitude excitation sinusoidal signal was 10 mV, and the frequency was scanned in a range of [10^−3^ Hz, 10^6^ Hz], as described by Gongi et al. [9].

Nyquist diagrams were recorded with increasing metal concentrations ranging from 10^−10^ M up to 10^−4^ M. All our measurements were performed in triplicates (*n* = 3) at a negative bias of −0.3 V. This value allows an improved definition in the Nyquist plot, being sufficiently low to reduce any corrosion phenomenon [4].

In this study, we used Nova 1.5 software (dedicated to impedance measurement), which is programmed to average the three replicates’ measurements (*n* = 3) and calculate their standard deviation. The equivalent circuit parameters for the electrolyte interface of the EPS gold sensor were interpreted after testing several models of the equivalent circuit.

The impedimetric responses of the EPS-functionalized electrodes to the metal ions Cd^2+^ and Hg^2+^ were investigated.

### 2.5. Love Wave Sensor

#### 2.5.1. Acoustic Love Wave Test Cell

The sensor platform used in this work was previously described by Tamarin et al. [21]. Briefly (Figure 4), the Love wave sensor unit (quartz + SiO_2_ guiding layer) consisted of two acoustic delay lines, with input and output interdigitated transducers each; one line was used as a reference and the other one for the measurement. The sensor unit was implanted in an experimental setup with a test cell ensuring electrical connections to the electronic readout circuit and a PDMS microfluidic chip localizing the aqueous sample on the surface of the sensor to prevent any contact with the electrical pads. A volume of 250 µL of EPS solution was injected into the PDMS chip and then dried for 24 h to obtain a thin EPS layer on the surface of the acoustic sensor.

#### 2.5.2. Acoustic Sensor Analytical Performances

Acoustic measurements for the electrical sensor were performed in air and water with a vector network analyzer (VNA; Anritsu MS4623B, Allen, TX 75013, U.S.A.). Results were expressed from real-time monitoring of the acoustic sensor transmission response (scattering parameter S21) in terms of the gain (insertion losses) and phase of the propagated acoustic wave between input and output IDTs. The frequency range of interest around the acoustic resonance was about 118 MHz.

#### 2.5.3. Real-Time Monitoring of Cellular Response to Heavy Metal

Similar measurements were carried out with a computer-controlled analyzer (Copper Mountain planar 304/1) to real-time monitor the response to heavy metals. The cadmium and mercury aqueous solutions at different concentrations were injected, and the relative insertion loss (dB) and phase (°) were determined and compared with the reference Love sensor response. All measurements were carried out in a controlled room to eliminate the effect of a variation in temperature or humidity. The responses of the Love wave sensor in air and in deionized water, as an adjacent medium, were considered references. Data represent the mean of three replicates (see Section 3.3.1 (Figure 13)).

#### 2.5.4. Sensitivity for Heavy Metals by EPS Acoustic Sensor

To go deeper into the analysis of the Love wave sensor response and determine the sensitivity, the insertion losses variation (Δ*Il*) and relative frequency shift at fixed phase ($\frac{\Delta f}{f_{\mathrm{ref}}}$) were calculated respectively according to Equations (1) and (2) as follows:(1)$$\Delta Il=Il_{\mathrm{meas}}-Il_{\mathrm{ref}}$$
(2)$$\frac{\Delta f}{f_{\mathrm{ref}}}=\frac{f_{\mathrm{meas}-}f_{\mathrm{ref}}}{f_{\mathrm{ref}}}$$

*Il_meas_* and $Il_{\mathrm{ref}}$ are respectively the measured insertion losses with increasing concentrations of heavy metals and in DI water at the resonance frequency, while $\;f_{\mathrm{meas}}$ and $f_{\mathrm{ref}}$ are respectively the measured frequency with increasing concentrations of heavy metals and in DI water at an equiphase point 0° near the resonance frequency.

### 2.6. Metal Solutions

Two types of metals, Cd (II) and Hg (II), were tested in this study. The stock solutions were prepared by dissolving Cd (Cl)_2_(H_2_O)_4_ and Hg (Cl)_2_ (Sigma-Aldrich, St. Louis, MO, USA) in distilled water. Aqueous metal solutions of concentrations ranging from 10^−10^ M up to 10^−3^ M were obtained by successive dilutions. All glassware was acid-washed before use to avoid the binding of metal.

### 2.7. Statistical Analyses

Statistical analyses were performed with SPSS ver. 20.0 professional edition. The impact of heavy metal on the variation in (dB) and (°) at all concentrations was evaluated with St-test’s *t*-test, and *p*-values of <0.05 were statistically significant.

## 3. Results and Discussion

The HPSEC elution profile (Figure 5) showed a single, symmetrical narrow peak, verifying the homogeneity of the EPS solution. Based on the calibration curve of the elution retention times of the standard dextran, the average molecular weight (Mw) of the EPS-based membrane solution was estimated to be 7.82 × 10^6^ g mol^−1^. This value is in the range of the EPS molecular weight reported in numerous studies [22]. In general, the variation in the molecular weight of EPS could be explained by differences in strains, fermentation factors, and EPS structures [23].

The zeta potential of the EPS membrane solution represents an index of intensity of electrostatic attraction between particles, showing that the *Graesiella* EPS were of anionic nature. The zeta potential was evaluated at −40 ± 2 mV. The negative charge may be due to the presence of anionic groups and to the presence of uronic acids [24]. Consequently, the EPS solution could be quite reactive with chemical species such as cadmium and mercury [25].

### 3.1. EPS-Functionalized Sensor Surface Characterization

#### 3.1.1. Atomic Force Microscopy (AFM) Analysis

The surface topography of the EPS-functionalized sensor is shown by the tapping mode 3D and 2D atomic force microscopy (AFM). The AFM profile displayed a crinkled and wrinkled structure with irregular blocks with a maximum height of 100 nm (Figure 6A). The 3D AFM images (Figure 6B) reveal that the EPS-functionalized sensor presented a pointed compact structural feature without protrusions. This structure can enlarge the active area of the sensor and help to immobilize selective materials including cadmium and mercury [26,27].

#### 3.1.2. Scanning Electron Microscopy of EPS-Functionalized Sensor

SEM micrographs yielded information about the membrane’s internal microstructures. The SEM top view and cross-section (Figure 7) confirmed a homogeneous character and revealed a microstructural lamellar arrangement of the EPS-functionalized membrane. Regarding the homogenous and compact structure, the surface morphology appeared without protrusions and with irregular blocks with similar features to that of native *Graesiella* EPS [24].

#### 3.1.3. FTIR Analysis

The presence of functional groups at the surface of the sensor was verified by FTIR measurements, as shown in Figure 8. FTIR spectroscopy showed little change in functional groups with respect to *Graesiella* sp. native EPS [24]. Broadband between 2900 and 3000 cm^−1^ is attributed to the stretching vibration (ν) of O-H or νC-H groups, characteristic of the hydroxyl and alkyl functionality of carbohydrates. The absorption observed at 1040 cm^−1^ could be related to the bending vibration (δ) of N-H and the νC-N, indicating the existence of amino acids from peptides/proteins. The small peak at 1229.82 cm^−1^ suggested the presence of sulfated groups, confirming the heterosulfated polysaccharides nature [28,29].

#### 3.1.4. X-ray Diffraction (XRD) Analysis

The X-ray diffraction (XRD) patterns of the EPS sensor membrane (Figure 9) exhibited numerous intense and sharp diffraction peaks ranging from 9 to 50°. Such a result indicates a highly crystalline nature, unlike that found in previous work [9], in which the amorphous nature characterized the sensor surface functionalized by cyanobacterial EPS. The crystalline nature is thought to give the EPS membrane a stronger interaction between the different structural components [16].

### 3.2. EPS-Functionalized Electrochemical Impedance Modeling

#### 3.2.1. Electrical Circuit Model

The electrochemical measurement is sensitive to changes at the interface of the electrode and the medium [4]. The EPSs gold sensor was modeled by an equivalent circuit (Figure 10) composed of two dipoles in series with electrolyte resistance (Rs). The first dipole (CPE EPS//Rm) models the electrochemical phenomena occurring at the membrane/electrolyte interface, with CPE EPS being the EPS membrane capacitance and Rm its resistance. The second dipole (CPE dL//Rct) describes the electron transfer impedance between the bulk and the electrode’s surface, with CPE dL the constant phase element of the charge transfer and Rct the electron transfer resistance. The same equivalent circuit was used to describe several biosensors, for example, those based on immobilized bacteria [3,5] or based on a monolayer EPS membrane [9].

The Nyquist diagrams (Figure 10) obtained for the gold electrode after the EPS deposition showed good agreement between the measured data and the fitting curves with chi-square values (χ^2^) of 0.02, indicating that this equivalent circuit is suitable and meaningful for this electrochemical system.

In the case of the two-dipole circuit, the total impedance of the constant phase elements ZCPE modeling the behavior of the interface is expressed by Equation (3), combining the CPE of the EPS membrane and that of the electrode surface [3,5,9]. Such an equation was used to decorrelate the impedance spectroscopy data parameters relative to each part of the equivalent circuit, as reported in Table 1.
(3)$$\mathrm{ZCPE}=\frac{1}{Q{\left(j\omega \right)}^{\propto }}=\frac{cos\left(\frac{\pi \alpha }{2}\right)}{Q\;\omega^{\alpha }}-j\;\frac{sin\left(\frac{\pi \alpha }{2}\right)}{Q\;\omega^{\alpha }}$$
where Q is a constant parameter, j is the imaginary number, ω = 2πf is the angular frequency, and α is a correction exponent (0 < α < 1).

#### 3.2.2. Effect of the Heavy Metal’s Concentration on the Impedance of the EPS-Functionalized Electrode

Electrochemical characterization of the EPS-functionalized sensors was investigated by plotting Nyquist diagrams resulting from electrochemical impedance spectroscopy (EIS) measurements of both mercury and cadmium at concentrations ranging from 10^−10^ M to 10^−4^ M (Figure 11).

The shape of Nyquist plots is related to the difference in the electrical signal formed due to the binding of heavy metal ions (Cd^2+^ and Hg^2+^) at the surface of the EPS-functionalized sensor. EPS acetate spectra (calibration curve) recorded before heavy metal injection showed the same pattern with high stability. An increase in the amplitudes of the Nyquist plots was observed in relation to an increase in metal concentration (Figure 11).

For the two heavy metals tested, the impedance parameters fitting the experimental data are shown in Table 1. Both the capacitance of the EPS membrane (CPE EPS) and that of the interface charge transfer (CPE dL) showed small variations with increasing metal concentration. Moreover, the quasi-stability of the correction exponent (α dEPS) and (α dL) near the unit value probably indicates no structural modification at the EPS membrane surface as well as at the electrode/membrane interface.The EPS-functionalized sensor showed high stability, a lower limit of detection, and affinity toward mercury and cadmium when compared with several gold electrodes [30,31].

EPS membrane resistance Rm significantly increased with increasing metal concentration (Figure 12). At low concentrations (10^−10^ to 10^−7^ M), the slope of the variation in Rm was more acute in the case of cadmium Cd^2+^ than with mercury Hg^2+^, indicating greater sensitivity toward cadmium ions. However, unlike Hg^2+^, in which no charge saturation was observed up to 10^−4^ M, a saturation of the sensor membrane occurred from a Cd^2+^ concentration as low as 10^−7^ M. Thus, obtained results confirm a higher range of mercury detection using the EPS sensor.

The value of Rm is correlated to the rate of exchange between the solution of metal cations and the negative charge of the surface of the EPS membrane. The rate exchange varies as a function of the metal species and its concentration in the parent solution, as well as its affinity with respect to one or other of the functional groups, including sulfate ones, of the membrane and their availability [32]. In consideration of the complexity of the EPS and its diversity in functional groups [14,15], further investigations are needed to understand these interactions in depth. In the case of other membranes, such as the demineralized lignite membrane, the exchange rate varied more rapidly with the concentration of Cd^2+^ than that recorded for Hg^2+^, and the adsorption kinetics of cadmium was almost twice that of mercury [33]. However, in other instances, Bhattacharjee et al. [34] demonstrated that Hg^2+^ ions are thiophilic and bind more easily to sulfate groups than Cd^2+^.

### 3.3. Love Wave Sensor Associated with EPS Membrane for Acoustic Detection of Heavy Metal in Liquid Medium

#### 3.3.1. EPS Membrane Influence

The impact of the EPS membrane on the performance of acoustic sensor platforms was investigated in terms of the gain, also referred to as insertion loss (dB), and phase (°) of the device transmission response using VNA.

Measurements were performed in air with the EPS-functionalized sensor (with EPS membrane) and compared with a noncoated sensor (without EPS membrane). The shift in insertion losses measured between the coated (−70.05 dB) and noncoated (−49.3 dB) Love wave sensors was −21 dB, revealing significant attenuation due to the presence of EPS (Figure 13). Commonly, a polymer membrane on the acoustic sensor surface impacts the wave propagation in a way that strongly depends on the material characteristics, especially its viscoelasticity.

Moreover, underwater acoustic measurement commonly reveals an additional attenuation of the acoustic wave in the order of −10 dB of losses [21]. In our case, and surprisingly, a good recovery of the “dB” responses with adjacent water can be observed (Figure 13), suggesting that EPS acts as a guiding layer for the acoustic wave, allowing enhanced propagation with water as a detection medium contrarily to air. Indeed, the acoustic attenuation shifted from −70.1dB (air) to −44.7 dB (water). This phenomenon could be attributed to the high potential of EPS to absorb and trap water molecules mainly through hydrogen bonding, modifying the mechanical properties of the material and thus reducing the acoustic wave attenuation [34], in agreement with crystalline behavior. Further investigations are required to deepen our understanding of the mechanisms involved. Nevertheless, *Graesiella* sp. EPS appears to be a potential candidate material for the coating of an acoustic sensor for heavy metal detection.

#### 3.3.2. Metal Detection by EPS-Functionalized Acoustic Sensor

The insertion losses at the operating frequency of the device were tracked to detect the impact of increasing heavy metal concentration, varying from 10^−11^ M to 10^−3^ M. The results presented in Figure 14 indicate that at the lowest mercury concentration tested ([Hg^2+^] = 10^−11^ M), no significant variation in insertion loss was observed (about −0.03 dB). However, while the concentration of mercury increased from 10^−10^ M up to 10^−3^ M, the gain peak values decreased distinguishably compared with the peak corresponding to the deionized water (DI, blank). The loss was −3.3 dB at 10^−10^ M and −10.8 dB at 10^−3^ M (Figure 14A). On the other hand, the equiphase frequencies were slightly shifted toward lower frequencies with increasing mercury concentration. The frequency value at 0° in phase changed from 117.28 MHz at 10^−10^ M to 117.25 MHz at 10^−3^ M, which led to a phase shift of 30 kHz (Figure 14B).

The gain (dB) and phase (°) spectra realized with the EPS-functionalized acoustic sensor for the detection of cadmium metal are illustrated in Figure 15. Similarly, the results clearly show that an increase in the cadmium concentration induces a decrease in the acoustic wave amplitude (Figure 15A). Again, the frequency also shifted toward low frequencies, as can be observed from the phase curves (Figure 15B), corresponding to a decrease in the acoustic wave velocity. No significant variation in the insertion loss was observed (about −0.1 dB) at a concentration of cadmium of 10^−11^ M. The additional insertion losses evaluated with reference to water were −0.7 dB at 10^−10^ M to −3.6 dB at 10^−3^ M. The frequency value at 0° in phase shifted from 117.558 MHz at 10^−10^ M to 117.553 MHz at 10^−3^ M, representing a six times lower detection limit for cadmium than mercury ions.

#### 3.3.3. Sensitivity toward Heavy Metals by EPS-Functionalized Acoustic Sensor

In this study, statistical analyses were performed using a student test to investigate the impact of the heavy metal concentration on the variation in dB and the frequency shift. As can be observed in Figure 16, the *Graesiella*-EPS-functionalized acoustic sensor exhibited an especially significant dB loss with mercury when compared with cadmium at all tested concentrations (*p* < 0.05). The Love wave electrode without EPS showed no sensitivity toward heavy metals, even at the highest concentration. In this regard, such EPS-functionalized sensors are able to detect cadmium and mercury with higher sensitivity toward mercury. That could be related mainly to the higher molecular weight of mercury when compared with cadmium. It was indeed shown that the adsorbed mass affects the response of the acoustic Love wave sensing and, hence, improves sensitivity [21,35].

## 4. Conclusions

This paper highlights the affinity of extracellular polymeric substances (EPSs) of thermophilic microalgae to bind heavy metals and induce surface-changing properties that can be detected by electrochemical impedance spectroscopy and/or acoustic wave sensing. The gold electrode and Love wave sensors used in this study for electrochemical and acoustic sensing showed god analytical performance and a low detection limit of 10^−10^ M. In summary, EPS biosensors could be a potential alternative tool for the detection of low concentrations of heavy metals and, in particular, aqueous Cd (II) and Hg (II). The advantages of using EPSs from microalgae as sensor bioreceptors lie in their ease of obtaining and in their natural biodegradable character. Based on the variety of functional groups, with differences between several EPS from different algae, it can be envisioned as a new basis for a multisensory device associated with the appropriate signal processing, especially based on new tools of recent advances in artificial intelligence and machine learning. However, further investigations are needed on the possible interactions in the presence of a mixture of different metals, as well as in the presence of other organic contaminants, if one aims for their use in water biomonitoring.

## Figures and Tables

**Figure 1 sensors-23-00803-f001:**
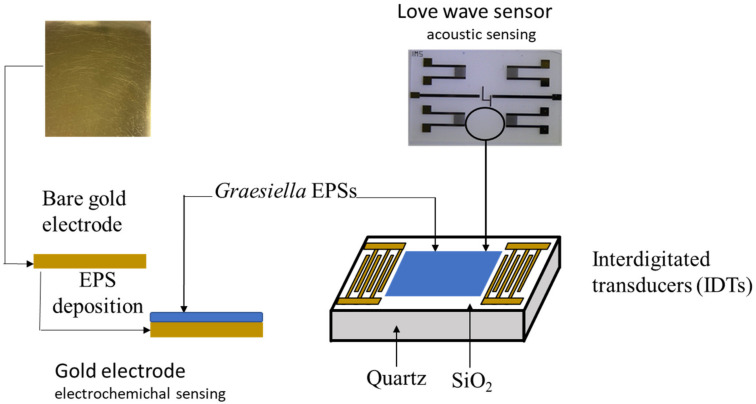
Scheme for the *Graesiella* EPS immobilization on the electrochemical sensor gold electrode (left) and the Love wave sensor surface (right).

**Figure 2 sensors-23-00803-f002:**
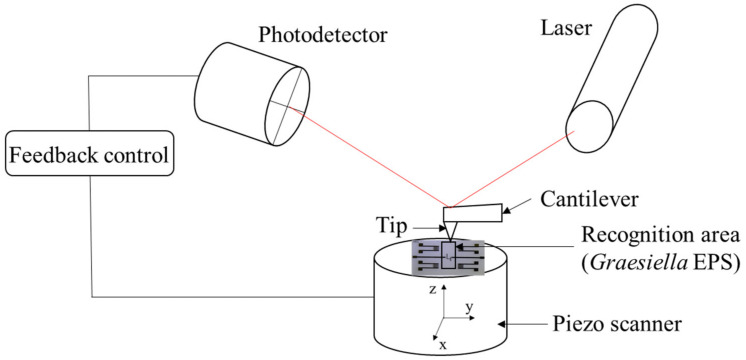
Schematic illustration of the Love wave EPS membrane AFM characterization.

**Figure 3 sensors-23-00803-f003:**
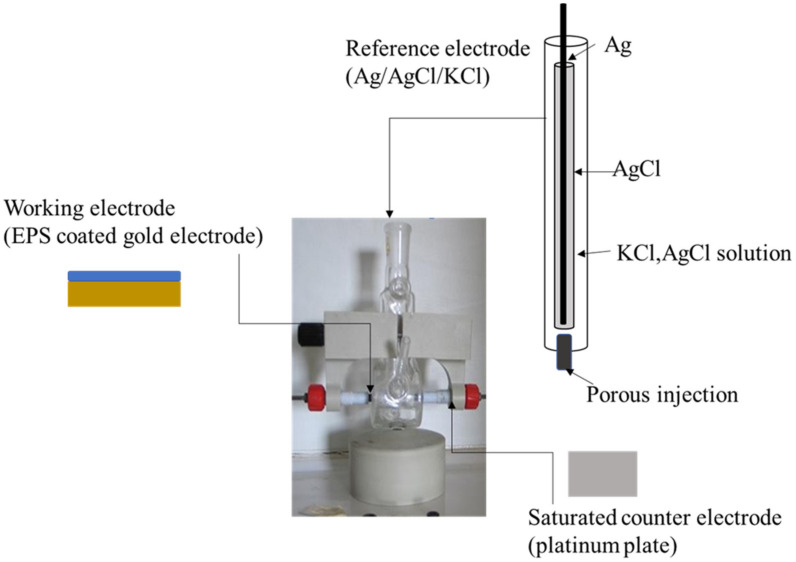
Experimental setup for electrochemical impedance measurements.

**Figure 4 sensors-23-00803-f004:**
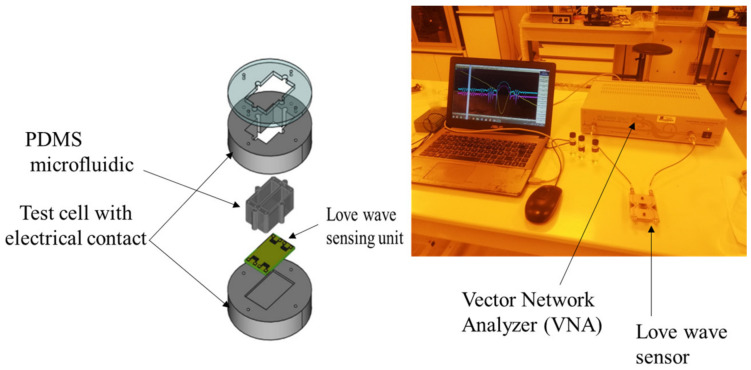
Experimental setup for acoustic sensing: Love wave sensor (left), sensing experimentation (right).

**Figure 5 sensors-23-00803-f005:**
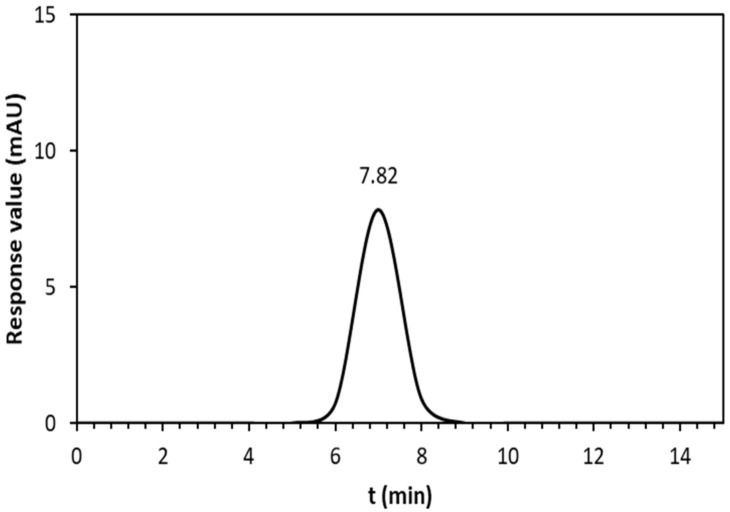
HPSEC chromatogram of EPS membrane solution; the vertical axis denotes the response values of the RI detector.

**Figure 6 sensors-23-00803-f006:**
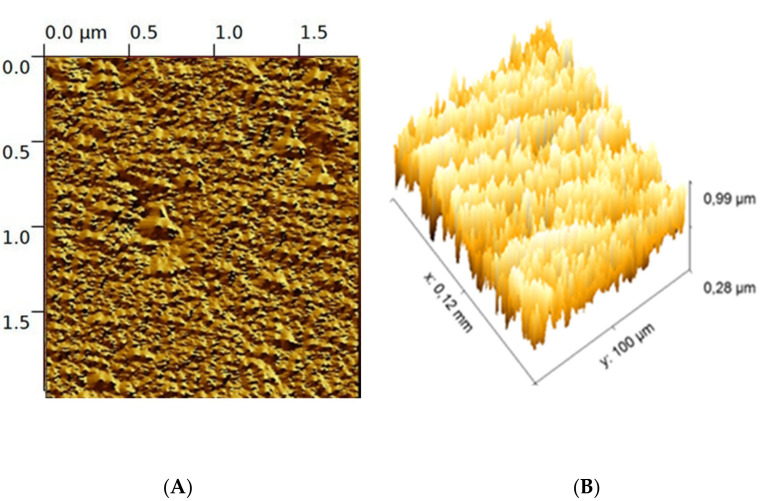
2D image (**A**), 3D (**B**) AFM image of *Graesiella* EPS-functionalized sensor.

**Figure 7 sensors-23-00803-f007:**
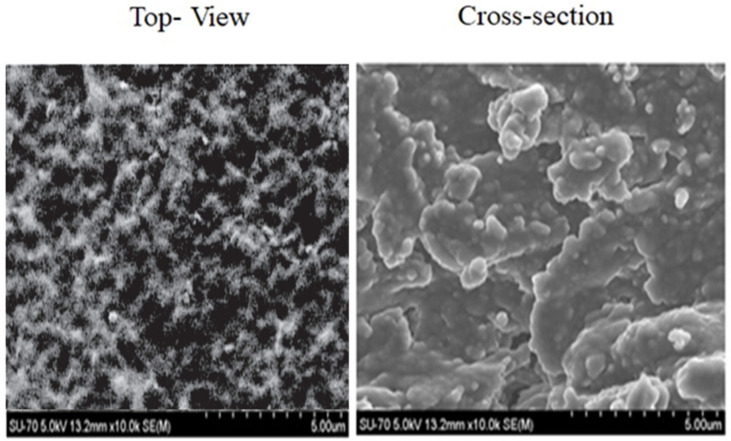
Scanning electron microscope micrographs (SEMs) of the top view and cross-section of the EPS-functionalized sensor observed for absolute pressure 60 Pa and accelerating voltage 5.0 kV.

**Figure 8 sensors-23-00803-f008:**
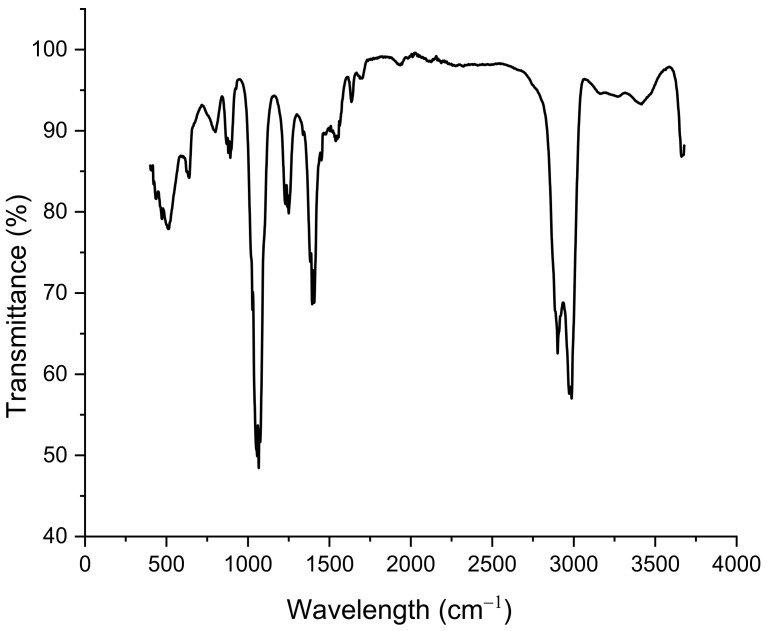
FTIR spectrum of EPS-functionalized sensor.

**Figure 9 sensors-23-00803-f009:**
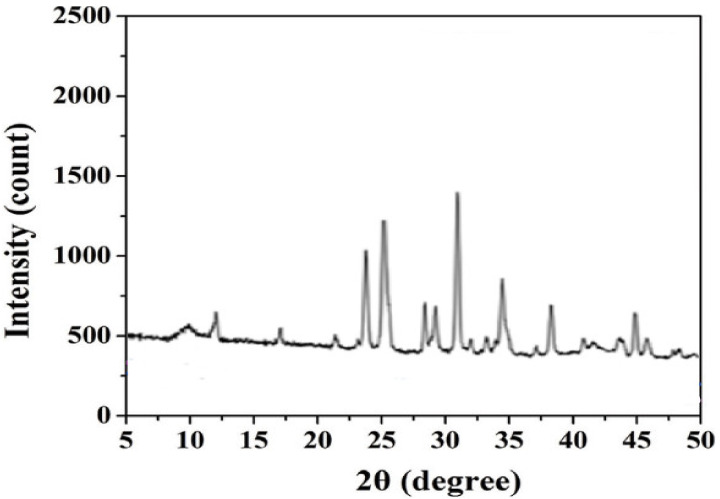
XRD profile of EPS-functionalized sensor.

**Figure 10 sensors-23-00803-f010:**
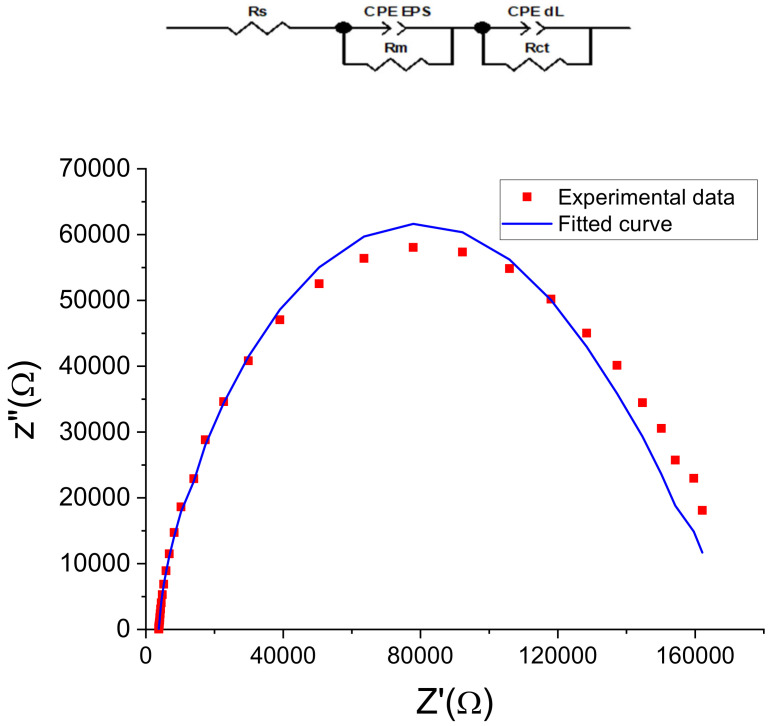
Nyquist plots (Z″ vs. Z′) of measured and modeled data of the EPS-functionalized gold electrodes. Dot curves represent the mean of measured values (in triplicate) calculated by Nova software, and continuous curves represent the corresponding fit.

**Figure 11 sensors-23-00803-f011:**
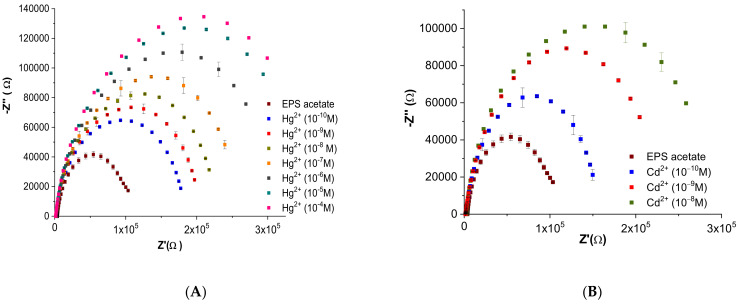
Nyquist plots of the gold electrodes coated with EPS membrane in acetate (10^−2^ M, pH 6.8) for a series of different concentrations of (**A**) mercury (Hg^2+^) and (**B**) cadmium (Cd^2+^) (values are presented as the mean of triplicate measurements (*n* = 3; ± SD).

**Figure 12 sensors-23-00803-f012:**
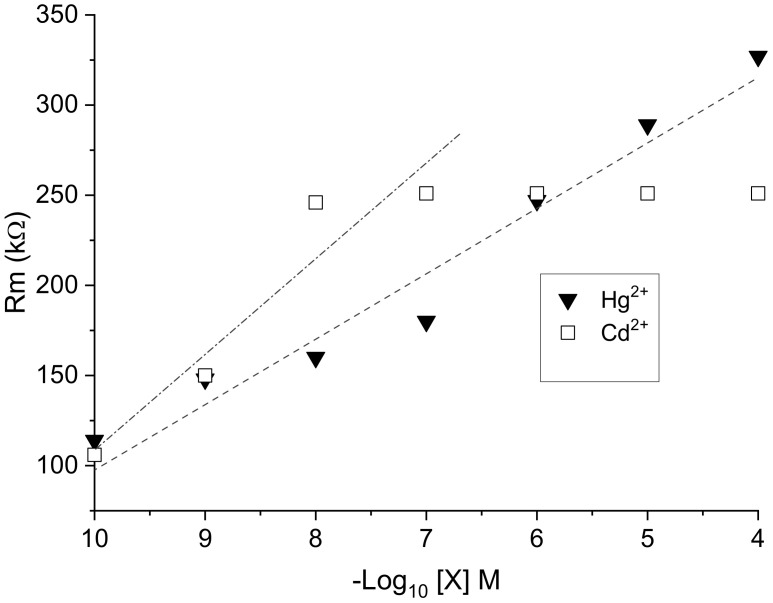
Variation in the membrane resistance Rm of EPS-functionalized electrochemical sensor as a function of −log[X] for X = Hg^2+^ and Cd^2+^.

**Figure 13 sensors-23-00803-f013:**
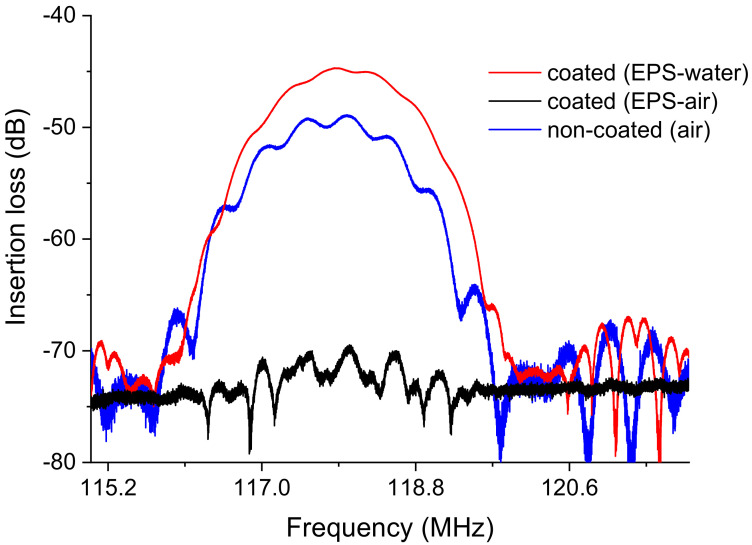
S21 transmission insertion losses of the Love wave sensor: noncoated with *Graesiella* sp. EPS in air and coated with *Graesiella* sp. EPS in air and water measured with VNA (vector network analyzer).

**Figure 14 sensors-23-00803-f014:**
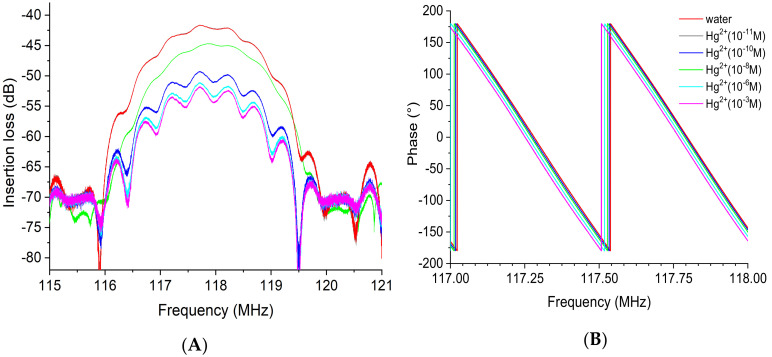
Superposition of the transmission response of the EPS-functionalized sensor with increasing mercury concentration from 10^−11^ to 10^−3^ M: (**A**) insertion loss or gain module and (**B**) phase. Both insertion losses (left) and phase (right) are plotted over a narrow frequency range centered on the acoustic resonance frequency.

**Figure 15 sensors-23-00803-f015:**
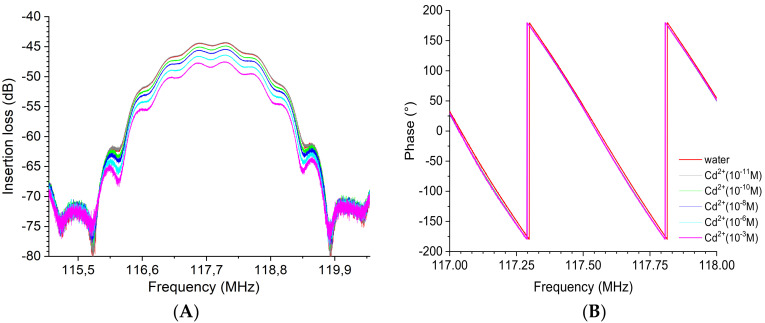
Superposition of the transmission response of the EPS-functionalized sensor for a series of increasing cadmium concentration from 10^−11^ to 10^−3^ M: (**A**) insertion loss or gain module and (**B**) phase. Both insertion losses (left) and phase (right) are plotted over a narrow frequency range centered on the acoustic resonance frequency.

**Figure 16 sensors-23-00803-f016:**
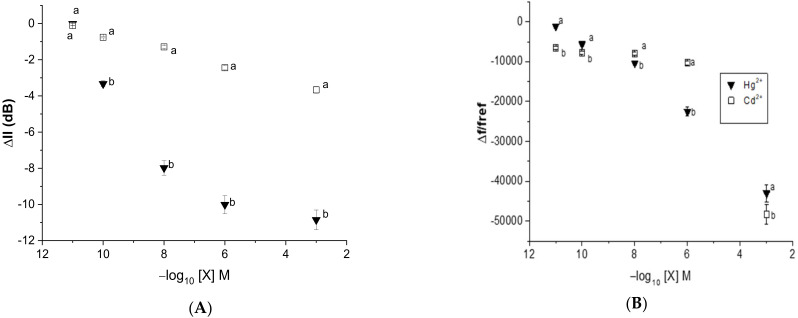
The average transmission response shift in (**A**) insertion loss (ΔIl) and (**B**) relative frequency shift (*Il_ref_*) vs. heavy metal concentration for the acoustic wave sensor coated with *Graesiella* sp. EPS. Values are presented as the mean of triplicate measurements (*n* = 3; ± SD).

**Table 1 sensors-23-00803-t001:** Parameters of the electrical circuit equivalent to the EPS-functionalized gold electrode for different concentrations of mercury (Hg^2+^) and cadmium (Cd^2+^); measurement was performed until saturation. Values are expressed as the mean of three replicates (*n* = 3) ± SD.

Hg^2+^ (M)	Rs (kΩ)	Rm (kΩ)	CPE EPS (µF)	α dEPS	Rct (kΩ)	CPE dL (10^−3^ µF)	α dL
10^−10^	1.80 ± 0.12	114.11 ± 2.21	2.59 ± 0.51	0.87 ± 0.01	66.94 ± 3.21	2.41 ± 0.09	0.84 ± 0.01
10^−9^	1.98 ± 0.32	148.62 ± 3.12	2.58 ± 0.43	0.87 ± 0.02	64.26 ± 2.45	2.20 ± 0.01	0.87 ± 0.02
10^−8^	1.97 ± 0.21	152.21 ± 3.15	2.70 ± 0.32	0.84 ± 0.02	59.04 ± 5.98	2.72 ± 0.07	0.94 ± 0.03
10^−7^	1.99 ± 0.32	168.31 ± 2.87	2.48 ± 0.21	0.87 ± 0.01	56.30 ± 4.21	2.11 ± 0.06	0.91 ± 0.02
10^−6^	1.98 ± 0.11	246.71 ± 4.21	2.38 ± 0.51	0.84 ± 0.02	53.50 ± 2.22	1.09 ± 0.04	0.96 ± 0.03
10^−5^	1.84 ± 0.22	289.09 ± 3.18	2.26 ± 0.02	0.83 ± 0.03	50.21 ± 2.31	0.89 ± 0.01	0.99 ± 0.04
10^−4^	1.18 ± 0.03	327.41 ± 3.41	2.01 ± 0.08	0.83 ± 0.05	42.40 ± 3.41	0.42 ± 0.04	0.99 ± 0.03
Cd^2+^ (M)							
10^−10^	1.74 ± 0.22	106.31 ± 1.32	1.82 ± 0.32	0.85 ± 0.01	60.79 ± 1.32	1.71 ± 0.02	0.90 ± 0.02
10^−9^	1.38 ± 0.21	150.22 ± 3.21	1.82 ± 0.08	0.89 ± 0.03	40.38 ± 2.43	0.91 ± 0.02	0.86 ± 0.02
10^−8^	1.41 ± 0.31	246.21 ± 1.52	1.72 ± 0.04	0.94 ± 0.04	35.31 ± 3.33	0.81 ± 0.04	0.86 ± 0.00
10^−7^ to 10^−4^	1.03 ± 0.51	251.31 ± 2.23	1.16 ± 0.21	0.99 ± 0.01	27.22 ± 2.98	0.92 ± 0.01	0.85 ± 0.01

Rs, solution resistance; Rm, resistance of the EPS membrane; CPE EPS, constant phase element of the EPS membrane; CPE dL, constant phase element of the interface; Rct, ion transfer equivalent resistance; α EPS and α dL are the correction exponent corresponding to the membrane and the interface, respectively.

## Data Availability

Not applicable.

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
