# Peer review of "Elaboration and Characterization of a New Heavy Metal Sensor Functionalized by Extracellular Polymeric Substances Isolated from a Tunisian Thermophilic Microalga Strain Graesiella sp."

_sensors, 2023, doi:10.3390/s23020803_

Round 1
Reviewer 1 Report (Previous Reviewer 1)
The authors promptly replied to all comments, clarified all doubts, completed the work with the missing information and corrected errors and inaccuracies. The new version of the work seems accurate to me, the only corrections to be made I think can be taken over by the staff that takes care of correcting the proofs, as they only concern the layout of the work. For me, it can be published in the present form.
Author Response
Thank you for your time.
Reviewer 2 Report (Previous Reviewer 2)
The authors revised their manuscript with careful response to my comments. So, this revised version could be accpetable.
Author Response
Thank you for your comments.
This manuscript is a resubmission of an earlier submission. The following is a list of the peer review reports and author responses from that submission.
Round 1
Reviewer 1 Report
Please see the attached file.

Reviewer 2 Report
This paper prepared an electrode functionalized by extracellular polymeric substances (EPS) isolated from a Tunisian thermo-15 philic microalga strain Graesiella sp as impedimetric and acoustic sensors for the detection of Hg2+ and Cd2+. The manuscript has the following problems.
1. There are many grammar errors throughout the manuscript. The writing of this paper should be corrected by a native English speaker.
2. In subsection 3.3.1, the discussion for Figure 10A is not necessary since this model is not successful.
3. In subsection 3.3.2, there is no any calibration curve for impedimetric detection of Hg2+ or Cd2+. So, it can not be said as an impedimetric sensor for Hg2+ and Cd2+ until the corresponding calibration curves are added and discussed. By the way, the impedimetric is influenced by only three different Hg2+ concentrations. How it can be used for Hg2+ detection?
4. The impedimetric and acoustic responses of other heavy metals should also be checked to discuss the interference of the sensors. The reasons for the selective responses of the sensors toward Hg2+ and Cd2+ should be explained.
5. The application of the proposed sensors for analyzing Hg2+ and Cd2+ in environmental water samples should be studied to demonstrate their applicability.
Reviewer 3 Report
The authors sucessfuly obtained and characterized two functionalized sensors with extracellular polymeric substances isolated from a Tunisian thermophilic microalga strain Graesiella sp. Sensitive membranes of the gold electrod and Love wave sensor have been characterized throught several techniques, i.e., the Fourier Transform Infrared spectroscopy (FTIR) and X-ray diffraction spectroscopy (XRD), atomic force microscopy (AFM) and scanning electron microscopy (SEM). The applied methods for sensor modification and characterization are detail provided.
However, a clearer manner of characterization results for each electrode should be provided. When the results are provided, the authors reffer to „sensor”, without mentioning which sensor’s analysis is given. Please provide a clearer manner of each sensor type characterization.
The importance of the substrate is not mentioned into the article. Please mention it.
For the electrochemical sensor, the gold electrode is used as support for extracellular polymeric substances (EPS) membrane. In literature, for electrochemical detection gold electrode has been reported able to detect heavy metals as mercury and cadmium. Please provide a comparison between the simple gold electrode and the EPS gold modified electrode for target heavy metals.
For the acoustic Love wave sensor you provided a comparison between unmodified and modified sensor. The comparison is provided in air. Having in view, the further application for the detection of heavy metals in water, please provide the Love wave sensor response unmodified and modified in water, and then at least the comparative response for one concentration of targeted heavy metals at modified and unmodified sensor.
Please pay attention to some minor English errors, e.g., Microscopy electronic balayage (MEB) (row 17-18).
I consider the paper could be accepted for publication after major revisions.